# Quantification of critical particle distance for mitigating catalyst sintering

Peng Yin[1,2,5], Sulei Hu[1,3,5], Kun Qian[1,3], Zeyue Wei[1,3], Le-Le Zhang[1,2], Yue Lin [1✉], Weixin Huang [1,3], Haifeng Xiong [4], Wei-Xue Li [1,3✉] & Hai-Wei Liang [1,2✉]

Supported metal nanoparticles are of universal importance in many industrial catalytic processes. Unfortunately, deactivation of supported metal catalysts via thermally induced sintering is a major concern especially for high-temperature reactions. Here, we demonstrate that the particle distance as an inherent parameter plays a pivotal role in catalyst sintering. We employ carbon black supported platinum for the model study, in which the particle distance is well controlled by changing platinum loading and carbon black supports with varied surface areas. Accordingly, we quantify a critical particle distance of platinum nanoparticles on carbon supports, over which the sintering can be mitigated greatly up to 900 °C. Based on in-situ aberration-corrected high-angle annular dark-field scanning transmission electron and theoretical studies, we find that enlarging particle distance to over the critical distance suppress the particle coalescence, and the critical particle distance itself depends sensitively on the strength of metal-support interactions.

[1] Hefei National Laboratory for Physical Sciences at the Microscale, University of Science and Technology of China, Hefei, China. [2] Department of Chemistry, University of Science and Technology of China, Hefei, China. [3] Key Laboratory of Surface and Interface Chemistry and Energy Catalysis of Anhui Higher Education Institutes, Department of Chemical Physics, University of Science and Technology of China, Hefei, China. [4] State Key Laboratory of Physical Chemistry of Solid Surfaces, College of Chemistry and Chemical Engineering, Xiamen University, Xiamen, China. [5] These authors contributed equally: Peng Yin, Sulei Hu. ✉email: linyue@ustc.edu.cn; wxli70@ustc.edu.cn; hwliang@ustc.edu.cn

Supported metal nanoparticles play a pivotal role in many industrial catalytic processes including the production of chemicals and fuels, automobile exhaust treatments, and fuel cells for clean energy technologies[1–3]. The performance of supported catalysts has a strong dependence on the metal particle size; the optimal particle size of <3 nm is generally required for increasing the specific surface areas of active metals and thus their mass-normalized activity[4]. However, when metal nanoparticle catalysts with small size are used in realistic reaction environments especially at high temperatures, they have strong tendency to undergo sintering (growth into larger particles) owing to the sharply increased surface energy with the decreased particle size. Such metal sintering inevitably leads to the loss of active surface area and thus to the catalysts deactivation[5–7]. Fundamental understanding of catalyst sintering is imperative to develop thermally-stable catalysts for sustainable and economically catalytic processes, particularly when considering the dwindling supplies of noble metals and increasing demand[8–11].

In the microscopic view, the metal sintering mechanisms can be classified into two types by specific transport pathways: particle migration and coalescence (PMC) and Ostwald ripening (OR)[12]. PMC involves particles migration under Brownian motion collide and consequent coalescence when two particles end up in close proximity; OR involves atom or molecular specie migration driven by reducing the chemical potential, in which atomic or molecular species are emitted from one particle, diffuse over the supports and attach to another particle, resulting in gradual growth of bigger particles and consumption of smaller particles. Accordingly, significant efforts have been devoted to developing sintering-resistant catalysts by spatially isolating active metal sites and thus hindering the metal specie migration[13,14], such as coating with inorganic shells[5,15], sandwiching between porous cores and shells[16,17], or confinement in the channels of zeolites[18,19], ordered mesoporous silica[20] and metal-organic frameworks[21]. Despite of promising approaches, these physical barriers often diminish the exposure of active metal surface and increase the mass-transfer resistance[13]. Besides, the elaborative geometric modification of the physical barriers may need tedious multistep processes and therefore are difficult to generalize.

Back to the sintering pathways, sintering happens when particles (via PMC mechanism) or atom/molecular species (via OR mechanism) stride across particle distance on supports to establish contact with each other[20,22]. Thus, in principle, enlarging the particle-to-particle diffusion distance by using high-surface-area supports and achieving uniform spatial distributions[23] of metal particles could slow down the metal sintering. The role of particle distance in catalyst sintering has been considered recently by a few groups but did not attract enough attentions[20,24–27]. Further, there is no report yet so far on the quantification of the particle distance over which the metal sintering can be significantly suppressed, though such fundamental knowledge is vital for developing sintering-resistant catalysts.

Here, we present the quantification of the critical particle distance for suppressing metal sintering on carbon supports. We adopt Pt nanoparticles supported on four commercial carbon black supports with specific surface areas ranging from 250 to 1500 $m^2 g^{-1}$ to construct model systems, which allows for precisely controlling the particle distance of various Pt/C catalysts by changing carbon black support, metal loading, and thermal treatment temperature. By the Pt/C model study, we can quantify a critical particle distance over which the Pt sintering can be significantly suppressed even up to 900 °C. By in situ aberration-corrected high-angle annular dark-field scanning transmission electron (HAADF-STEM) and theoretical studies, we identify that the particle coalescence sintering can be suppressed by enlarging the particle distance to over the critical value. It is found that the

quantified critical particle distance is highly dependent on the strength of metal-support interactions. The critical distance can be shortened and the critical loading can be improved through strengthening the metal-support interaction, for example, by doping the carbon supports with sulfur. We also demonstrate the implementation of the critical distance concept in catalytic propane dehydrogenation.

## Results

**Quantification of critical particle distance.** Direct measurement of the particle distance by two-dimensional projection using electron microscopy is challenging on account of the three-dimensional complexity of real supports. Hence, we resorted to the equation that was firstly described by Mayrhofer et al. to estimate the average particle distance by assuming the perfectly uniform particle size and equidistant distribution of particles (Eq. 1)[25]:

$$d = \sqrt{\frac{\pi}{3\sqrt{3}} \cdot 10^{-3} \cdot \rho_{pt} \cdot \left(\frac{100 - W_{pt}}{W_{pt}}\right) \cdot A_s \cdot r^3} - r \quad (1)$$

where $d$ is the average particle distance (nm), $\rho_{Pt}$ is the density of bulk platinum (21.45 g cm$^{-3}$), $W_{Pt}$ is the Pt loading (wt %), $A_S$ is the specific surface area of the supports (m$^2$ g$^{-1}$) and $r$ is the particle size in diameter (nm). Therefore, the rate of sintering, which depends on the particle distance, is influenced by the metal loading and the specific surface area of supports for a given particle size. As illustrated in Fig. 1, $d$ stands for the particle-to-particle distance of neighboring two particles, while $d_c$ denotes the critical particle distance over which the metal sintering would

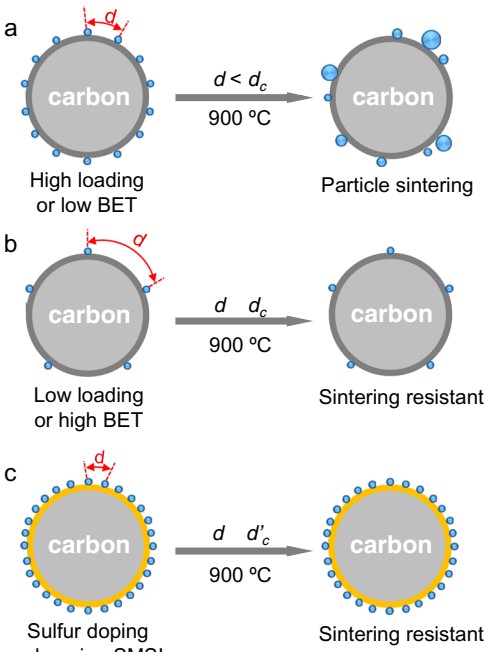

**Fig. 1 Schematic illustration of mitigating sintering by controlling particle distance. a** High metal loading or use of low-surface-area supports results in a shorter particle distance than the critical distance (that is, $d < d_c$), leading to severe sintering. **b** Enlarging the particle distance to above $d_c$ (that is, $d \geq d_c$) by lowering metal loading or using high-surface-area supports results in sintering-resistant catalysts. **c** Strengthening the interaction between metal and carbon supports by sulfur doping can shorten the critical particle distance and thus achieve sintering-resistant catalysts with high metal loading.

be mitigated at a given temperature. For the catalyst system with $d < d_c$, that is, high metal loading or use of low-surface-area supports, the particles presuming in similar average size may readily encounter to each other and grow into large particles (Fig. 1a). On the contrary, the use of high-surface-area supports or lowering the metal loading can enlarge the particle distance ($d \geq d_c$) and will mitigate the sintering (Fig. 1b).

For quantitatively corroborating the concept, we constructed Pt/C model systems by selecting four commercial carbon black supports with various specific surface areas to study the metal sintering (Supplementary Fig. 1), including Vulcan XC-72R (XC-72R, 250 $m^2 g^{-1}$), Ketjenblack EC-300J (KJ300J, 800 $m^2 g^{-1}$), Ketjenblack EC-600J (KJ600J, 1398 $m^2 g^{-1}$), and Black Pearls 2000 (BP2000, 1405 $m^2 g^{-1}$). We employed the industrially relevant method for the catalyst preparation, that is, wet-impregnation of supports with common metal precursors (i.e., $H_2PtCl_6$) followed with $H_2$-reduction at 300 °C. The sintering experiments were then carried out in the sintering-promoting reduction atmosphere (5.0 vol% $H_2$/Ar) at gradually rising temperature in the range of 300–900 °C for 120 min. We postulate that the average particles size of Pt retained below 3 nm after sintering test at 900 °C for 120 min as the indicator of sintering resistance, which is the desirable particle size for many catalysis applications[28–30].

We first screened the Pt loading for each support to find the critical loading without significant sintering observed, that is, the average particles size retained below 3 nm under maximum loading after sintering test. The particle size change of the Pt/C catalysts after each sintering test was analyzed by high-angle annular dark-field scanning transmission electron (HAADF-STEM), X-ray diffraction (XRD), and CO-stripping measurements. Specifically, the critical Pt loading was found to be 3, 8, 15, and 25 wt% on XC-72R, KJ300J, KJ600J, and BP2000, respectively. ICP-AES and $N_2$ sorption analyses revealed no change of the Pt content, pore volumes, specific surface area of these Pt/C catalysts upon high-temperature treatments, indicating the highly thermal stability of the commercial carbon black supports even with the presence of Pt nanoparticles (Supplementary Figs. 2 and 3). HAADF-STEM observations showed that the Pt nanoparticles at the critical loading homogeneously distributed over the carbon supports, without any obviously large particles or aggregations in all the sintering tests (Supplementary Fig. 4). In marked contrast, when the Pt loading increased by only 2–5 percentage points on the basis of critical loading for each support, we observed many abnormally large particles after the same sintering test, indicating that the critical loading on these supports have been achieved by decrease of the particle distance. The corresponding particle size distribution further clearly shows that once the Pt loading on each support exceeded the corresponding critical value, Pt nanoparticles would suffer sintering severely (Fig. 2a–d). We summarized the average particle sizes of the Pt/C catalysts with the critical loading after sintering tests at various temperatures in Fig. 2e. Remarkably, the average particle size for all catalysts was retained below 3 nm even after the sintering tests at high temperature up to 900 °C, though it slightly increased with the temperature.

To eliminate the possible illusion caused by the very local HAADF-STEM observations, we further estimated the average particle size of Pt by the Debye-Scherer equation based on the full-width at half-maximum of XRD patterns (Supplementary Fig. 5). We also found the loading-dependent sintering behavior by XRD: the low-loading samples exhibited remarkable sintering resistance (that is, the average particle size was kept below ~3 nm after sintering test at 900 °C for 120 min), while the high-loading samples underwent severe sintering with significant particle growth up to 10 nm (Fig. 2f). Further, we measured the

electrochemically active surface area (ECSA) for the Pt/C catalysts by the electrochemical CO-stripping technique, which is a reliable approach to assess the surface area of Pt/C catalysts via electrochemical oxidation of an adsorbed CO monolayer over Pt surface[31]. With the treatment temperature increased, the ECSA of different Pt/C system was gradually decreased and remained at higher levels equivalent to about 3 nm for the 900 °C sample. Once exceeding the critical Pt loading on each carbon support, we observed a significantly reduced ECSA for all the samples at 900 °C, indicating severe sintering and greatly losing of Pt active sites (Supplementary Fig. 6). The CO-stripping, XRD, and microscopic results were well consistent with each other; all the data together definitely demonstrated that sintering-resistant catalysts could be easily achieved just by controlling the metal loading on various carbon black supports with surface areas varying largely from 250 to 1500 $m^2 g^{-1}$ (Fig. 3a).

To identify the inherent parameter that governs the above loading-dependent sintering behavior of the Pt/C catalysts, we applied Eq. 1 to our experimental results for calculating the particle distance. Interestingly, we found a general critical particle distance to be of around 30 nm for XC-72R, KJ300J, and KJ600J, over which the Pt sintering could be mitigated greatly up to 900 °C. The critical particle distance of Pt nanoparticles on BP2000 was estimated to be a lower value of 23 nm, which is owing to the confinement effects induced by the numerous micropores in BP2000 (see below for more discussion on this issue). We further shortened the critical particle distance and thus improved the critical loading by strengthening the interaction between metal and carbon supports through the sulfur doping (Fig. 1c). Sulfur has a strong trend to form covalent bonds with Pt according to the soft acid–soft base interaction principle. We have previously demonstrated that mesoporous sulfur-doped carbons had strong capability for fixing metal atoms or nanoclusters owing to the strong chemical/electronic interaction between Pt and sulfur atoms doped in carbon matrix[32,33]. We here synthesized sulfur-doped BP2000 (S-BP2000) by carbonization of molecular precursors (2,2′-bithiophene) on the surface of BP2000[34,35]. S-BP2000 possesses a lower surface area (768 $m^2 g^{-1}$) compared to pristine BP2000, as the newly generated carbon derived from molecular precursors may block up a large number of pores of BP2000 (Supplementary Fig. 1). X-ray photoelectron spectroscopy analysis confirmed that the sulfur doping on the BP2000 surface with an initial content of 3.20 at% decreased gradually to 1.02 at% after the thermal treatment (Supplementary Fig. 7). We further investigated the change of sulfur doping upon the thermal treatment by energy dispersive spectroscopy (EDS) elemental mapping. Interestingly, we observed the spatial overlapping of Pt and residual sulfur elements (Supplementary Fig. 8), which strongly suggested the key role of the doped sulfur atoms as anchoring sites for Pt nanoparticles even at high temperature[36,37]. We also observed a significant 0.3 eV shift of the Pt 4f peak to a lower bind energy for Pt/S-BP2000 compared to Pt/BP2000 (Supplementary Fig. 9), again confirming the formation of the interfacial Pt-S bonds with electron donation from S-BP2000 to Pt[32]. Even though the lower surface area, the critical Pt loading on S-BP2000 could reach 35 wt% (Fig. 3a and Supplementary Fig. 10), indicating the effectiveness of strong Pt/S-BP2000 interaction for enhancing the sintering resistance. Accordingly, the critical particle distance was further shortened to 15 nm on the S-BP2000 support (Fig. 3b). It is worth highlighting that improving of Pt loading capacity on carbon supports is crucial for fuel cell applications by thinning the catalyst layer and thus enhancing the fuel cell performance[38]. Also, we have combined the concepts of critical particle distance and strong metal-support interaction to prepare other high-metal-loading sintering-resistant catalysts on the S-BP2000 support, including Ru, Rh, and Ir (Supplementary Fig. 11).

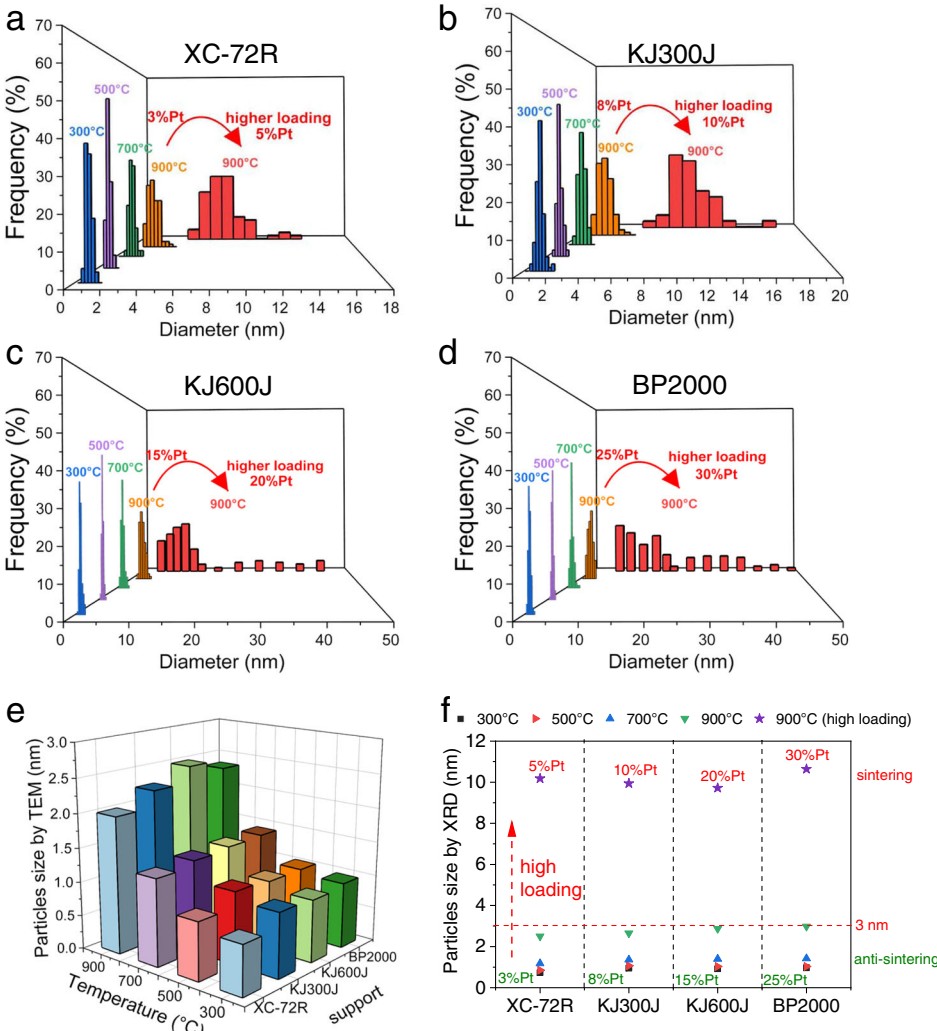

**Fig. 2 Particle size evolution of Pt/C catalysts during sintering tests. a–d** Particle size distribution of the Pt/C catalysts after sintering tests at different temperatures in 5% $H_2$/Ar for 120 min, clearly showing the strong loading-dependent sintering behavior for each carbon support. **e** Average particle size of the Pt/C catalysts with corresponding critical loading for each support, indicating the sintering resistance of the catalysts. The particle size was estimated from electron microscopy results. **f** Average particle size of the Pt/C catalysts estimated from XRD results, confirming the loading-dependent sintering behavior.

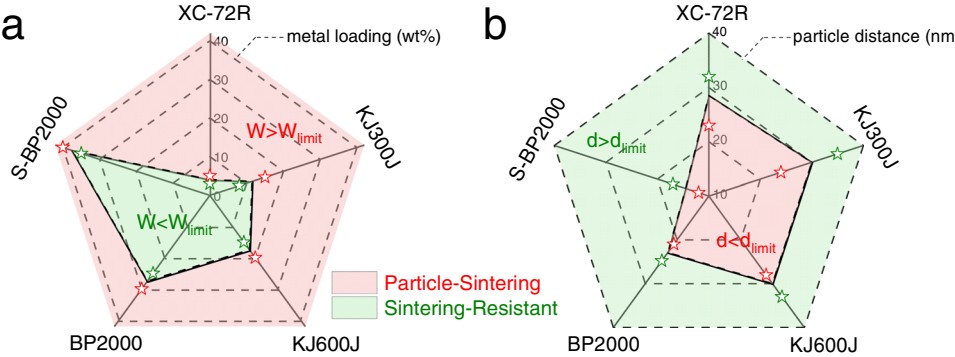

**Fig. 3 Critical Pt loading and particle distance. a** Critical Pt loading of the Pt/C catalysts for mitigating catalyst sintering. **b** Critical particle distance of the Pt/C catalysts for mitigating catalyst sintering.

**In situ aberration-corrected HAADF-STEM studies**. To fundamentally understand the reasons behind the influence of particle distance on sintering of Pt/C, we carried out the in situ aberration-corrected high-angle annular dark-field scanning transmission electron (HAADF-STEM) studies, which allow us to

track individual nanoparticles/atom species during heating and thus to identify the dominant sintering path involved in the Pt/C catalysts at atomic scale[39]. Considering the different conditions between vacuum and $H_2$/Ar, we carried out the sintering experiments in tube furnace under vacuum before in-situ STEM

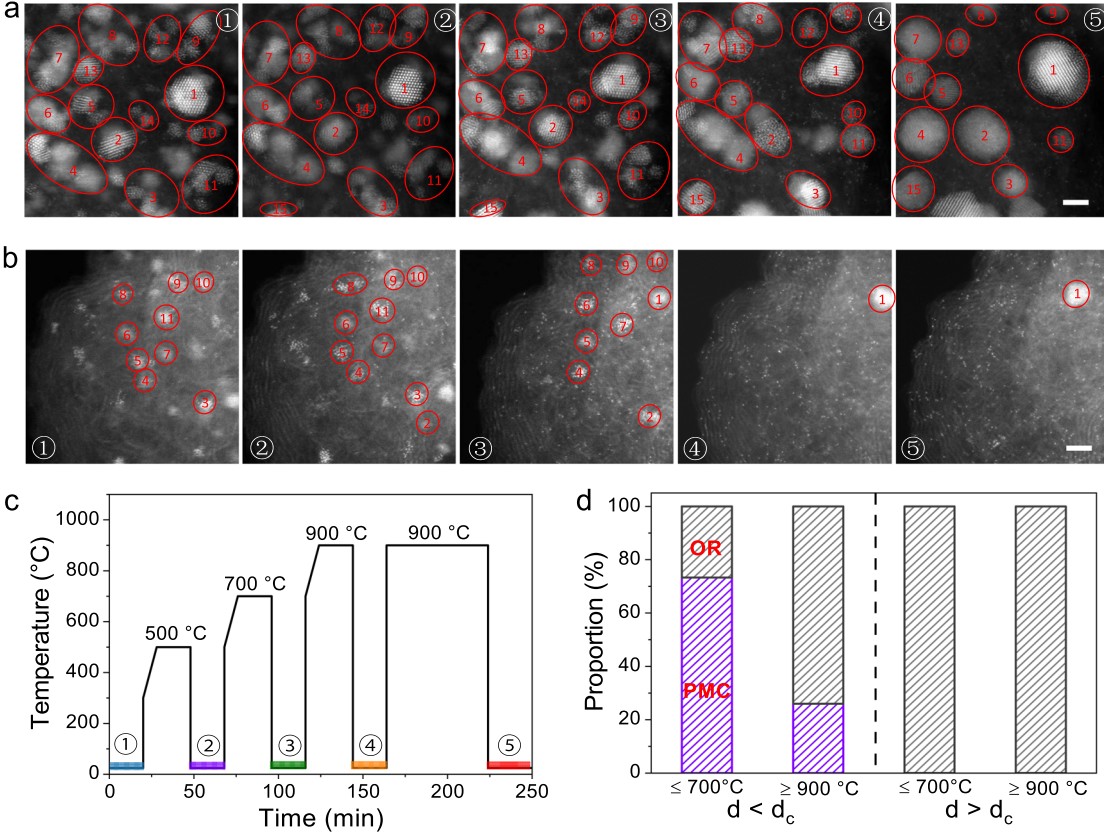

**Fig. 4 In situ sintering HAADF-STEM studies. a, b** In situ aberration-corrected HAADF-STEM images of 10%Pt/XC-72R (**a**) and 1%Pt/XC-72R (**b**) at different stages shown in (**c**). Scale bar, 2 nm. **c** Temperature-time program of the sintering protocol. HAADF-STEM observations were performed when the samples were quenched to room temperature (100 °C per second) after each stage of in situ heating. **d** Proportion of OR and PMC mechanisms identified for the short (10%Pt/XC-72R, $d < d_c$) and long (1%Pt/XC-72R, $d > d_c$) particle distance catalysts at the early stages (≤700 °C) and later stages (≥900 °C), respectively.

test. In vacuum, we observed similar particle distance dependent sintering behavior in 5% $H_2$/Ar: the short particle distance catalyst (10% Pt/XC-72R) was prone to severe sintering, while the long particle distance catalyst (10% Pt/ BP2000) exhibited a superior sintering-resistance (Supplementary Figs. 12). We prepared Pt/XC-72R at 300 °C with the loading of 1.0 wt% and 10.0 wt % as starting point for in situ HAADF-STEM studies, which represent the long and short particle distance sample, respectively. As expected, 10.0 wt% Pt/XC-72R showed much higher particle density and shorter particle distance than that of 1.0 wt% Pt/XC-72R (Fig. 4a, b, Supplementary Figs. 13 and 14). To eliminate the possible influence of electron beam particularly at high temperature[40], after each stage of in situ heating we quenched the hot sample to room temperature (100 °C per second) before the HAADF-STEM observations. The detailed in situ heating protocol was shown in Fig. 4c and five tagged stages were selected for the HAADF-STEM observations. We randomly observed several regions for each sample to follow the evolution of particle positions and sizes at the same area during different sintering stages (Fig. 4a, b). To exclude the possible "knock-on" damage at 200 kV, before individually tracking fixed area (Fig. 4a, b), we have also selected several other areas to compare the changes before and after heating (Supplementary Figs. 13-17), which were not disturbed by the "knock-on" damage from electron beams.

For the high-loading Pt/XC-72R (that is, $d < d_c$), in all the stages we could easily discern that the coalescence sintering path, that is, many particles with similar size gradually fused into a large particle (Fig. 4a and Supplementary Fig. 13). At the two later

stages of ≥900 °C, we meanwhile observed that numerous single atoms emerged over the carbon supports and some particles disappeared in some areas and reappeared in new places (Fig. 4a and Supplementary Fig. 13), which all verified a clear pattern of Ostwald ripening sintering path. To quantitatively analyze the dominant sintering path, we further split the in-situ heating experiment into the early three stages of ≤700 °C and the two later stages of ≥900 °C for counting the instances of coalescence and Ostwald ripening, respectively. In the three early stages, we counted three regions of 131 nanoparticles, including 35 instances of Ostwald ripening versus 96 instances of coalescence (Fig. 4d and Supplementary Table 1). While in the two later stages, the number of particles showed a sudden drop to 52 in later stages, with 37 instances of Ostwald ripening versus 15 instances of coalescence counted.

The different dominant sintering path in the early (PMC-dominated) and later (OR-dominated) stages of the high-loading Pt/XC-72R catalyst could be explained by the changed particle distance along the sintering proceeding. At the early sintering stages of the high-loading Pt/XC-72R catalyst, Pt nanoparticles are located in close proximity and, therefore, are more susceptible to sinter via migration and coalescence[8]. With the sintering proceeding, the nanoparticle coalescence would lead to the decrease of particle density and the enlarged particle distance, which, in turn, slowed down the sintering by coalescence. On the other hand, the wide particle size distribution caused by coalescence could intensify differences in chemical potentials of Pt particles and thus aggravate the Ostwald ripening during the later sintering stages at high temperatures[41].

In the case of low-loading Pt/XC-72R (that is, $d > d_c$), we found that many particles gradually became smaller with the temperature rising and disappeared completely at 900 °C. Furthermore, particles emerged in new locations and grew into large ones with a plenty of single atoms surrounding (Fig. 4b and Supplementary Fig. 14), indicating the typical Ostwald ripening mechanism involving the mass transfer from the smaller to the larger particles via mobile of atomic species along the two-dimensional support surface. We did not find any instances of particle coalescence in all the sintering stages for the low-loading Pt/XC-72R sample (Fig. 4d). Large region HAADF-STEM observation with more than 500 particles also gave the same conclusion (Supplementary Fig. 15 and Video 1). It is worth noting that in some regions there are only single atoms without nanoparticles, suggesting the required long-distance transport of atomic species for the OR-type sintering. Similarly, for the long particle distance Pt/BP2000 catalyst (10 wt% Pt, $d > d_c$), we also identified that the coalescence sintering path was suppressed completely and that Ostwald ripening dominated the Pt sintering under high temperatures (Supplementary Figs. 16 and 17). Meanwhile, we supplemented the only time-resolved STEM experiments with only electron beam treatment for 30 min (Supplementary Fig. 18 and Videos 2, 3) and then started heating at 900 °C. We also performed the only heating STEM experiments, where the electron beam was closed during heating and only open to observe changes after the end of heating (Supplementary Figs. 19). Only the "knock-on" damage of electron beam cannot change the particle state, but the thermal energy can induce changes in particles. These results suggested that the long particle distance sample was prone to Ostwald ripening via atom species only derived by the thermal energy force, rather than the "knock-on" damage from electron beam.

Overall, the in situ aberration-corrected HAADF-STEM studies revealed that the particle coalescence dominated the sintering of Pt/C catalysts with short particle distance along with Ostwald ripening at later sintering stages, and, more importantly, that enlarging of particle distance could suppress the particle coalescence sintering and meanwhile slow down the Ostwald ripening.

**Computation study**. To reveal the underlying physics of the particle distance dependent sintering of the Pt/C catalysts, we performed the systematical computational studies. According to the above in situ aberration-corrected HAADF-STEM studies, the Pt sintering is found to proceed majorly via the PMC path, which depends not only on the particle spatial and size distribution but also the metal-support interaction (MSI). The corresponding rate is determined essentially by particle diffusion coefficient on supports as derived in our previous work[42]:

$$D_p(R) = \frac{K_\alpha}{R^4} \exp\left[\frac{\Delta\mu(R)}{k_B T}\right] \exp\left[-\frac{E_{act}^m - S^m E_{adh}}{k_B T}\right] \quad (2)$$

where $R$ is the radius of curvature of the supported particle, $K_\alpha$ is the structure factor related to the contact angle $\alpha$ of metal particle with the support (see more details in Supplementary Materials), $k_B$ is Boltzmann's constant, $T$ is temperature. $E_{act}^m$ is the self-activation energy of metal atoms that characterizes its formation and migration on the surface of nanoparticle and is considered as one-third of the cohesive energy of bulk metal (1.93 eV for Pt)[43]. $S^m$ is the area of the Pt atom (6.17 A$^2$)[43], and $E_{adh}$ is the interfacial adhesion energy between metal and support and is closely related to $\alpha$ according to the Young-Dupre' equation[43]

$$E_{adh} = -\gamma_m(1 + \cos\alpha) \quad (3)$$

where $\gamma_m$ is the experimental surface energy of the Pt bulk (0.155 eV/A$^2$)[43]. $\Delta\mu(R)$ is the chemical potential of metal atoms. $\Delta\mu(R)$

in supported nanoparticles with respect to the bulk counterpart can be calculated approximated by Gibbs–Thomson (G–T) relation[23]

$$\Delta\mu(R) = \frac{2\Omega\gamma_m}{R} \quad (4)$$

where $\Omega$ is the volume of Pt atom (15.33 A$^3$)[43]. For small nanoparticles, $\gamma_m$ becomes size dependent due to the increase of low coordination sites, as shown in Supplementary Fig. 20. When a metal particle with a given volume $V_0$ lands on the support of interest, it tends to adopt into a truncated sphere shape with $\alpha$, and the corresponding $R$ is determined by

$$R = \sqrt[3]{3V_0/4\pi\alpha_1} \quad (5)$$

where

$$\alpha_1 = \frac{(2 - 3\cos(\alpha) + \cos^3(\alpha))}{4} \quad (6)$$

We performed the high resolution transmission electron microscopy (HRTEM) observations to roughly extract the corresponding $\alpha$ of Pt nanoparticles on carbon supports and (Supplementary Fig. 21). By statistically analyzing the HRTEM images, the average $\alpha$ was derived for each sample and plotted in Supplementary Fig. 22. It was found that XC-72R has the largest $\alpha$ of 125.4° representing a relatively weak MSI, followed by KJ300J (124.1°), KJ600J (118.9°), BP2000 (116.2°), and S-BP2000 (108.7°) representing a relatively stronger MSI. Corresponding $E_{adh}$ between Pt and five carbon supports based on the $\alpha$ data were calculated. Based on the extracted $\alpha$ and taking into account of the size effect on $\gamma_m$, $\Delta\mu$ of Pt atoms on these carbon supports was calculated and plotted with respect to the particle in Fig. 5a. At a given particle size, the Pt nanoparticles on XC-72R was found to have a highest $\Delta\mu$, whereas have a lowest one on S-BP2000 due to the strengthened MSI. We then calculated $D_p$ of Pt nanoparticles versus the particle size on different carbon supports at 900 °C, as shown in Fig. 5b. For small 2.8 nm Pt of interest, the calculated $D_p$ is as high as 24.3 nm$^2$ s$^{-1}$ on XC-72R, but only 1.6 nm$^2$ s$^{-1}$ on S-BP2000. Such difference by a factor of fifteen along with the change in $\alpha$ of 20° is significant, indicating the great influence of MSI on nanoparticle diffusion. Note that $D_p$ decreases dramatically with the particle size, due to the inverse fourth power relationship on $R$ (Eq. 2).

For a time period of $\tau$, the spatial dimension of the Pt nanoparticle diffusion via Brownian motion can be measured by a characteristic diffusion length $L$:

$$L = [D_p(\alpha)\tau]^{1/2} \quad (7)$$

As shown in Fig. 5c, for $\tau = 2$ h, $L$ is insensitive to $\alpha$ at a relatively low temperature of 700 °C. When increasing the temperature up to 900 °C, $L$ increases considerably and becomes sensitive to $\alpha$, and specifically, 201 nm on XC-72R (125.4°) versus 52 nm on S-BP2000 (108.7°). Small $L$ on S-BP2000 than that on XC-72 R implies low probability for two particles to collide and merge into a larger particle, corroborating well with the trend behavior of experimentally extracted critical particle distance. Supposing the support with an area of $\pi(L/2)^2$ only supports one particle, the critical loading of the supported Pt nanoparticles with a volume of $4\pi\alpha_1 R^3/3$ can be estimated by:

$$\text{Loading} = \frac{16S_m R^3 \alpha_1 m_{Pt}}{3L^2\Omega}, \quad (8)$$

for a given mass specific surface area $S_m$ of the support, where $m_{Pt}$ is Pt atom mass (3.24 × 10$^{-22}$ g/atom)[43]. For 2 nm Pt dispersed on five carbon supports and aging at 900 °C for 2 h, the maximum loadings are calculated and compared with the experiments. As

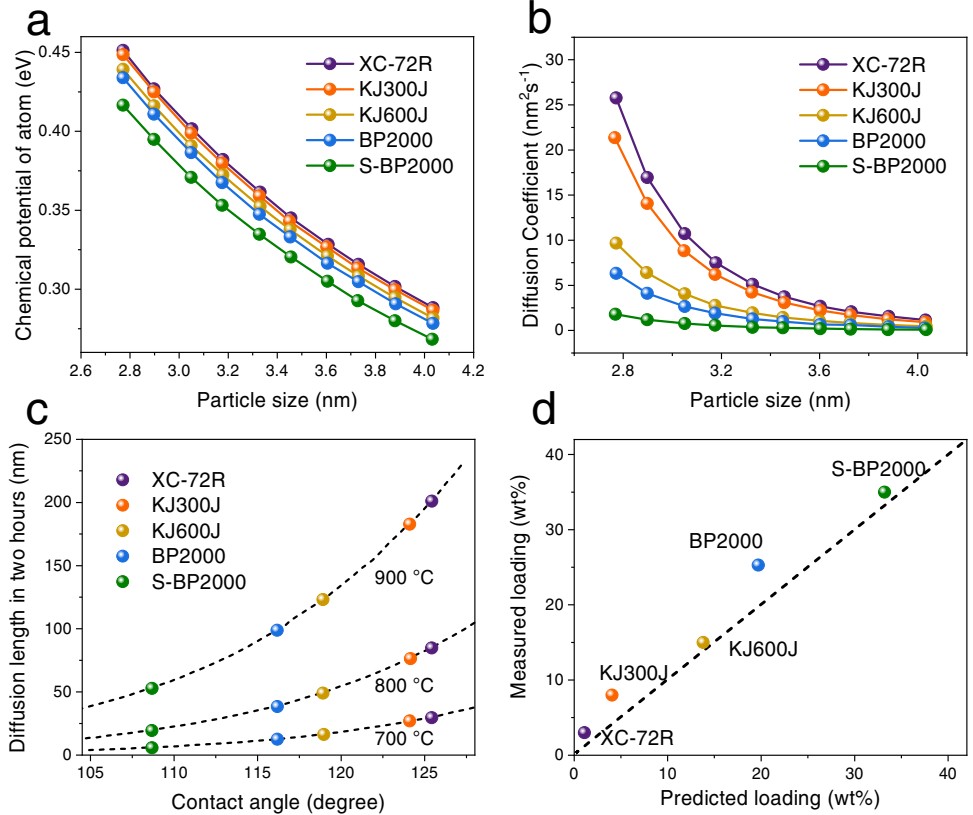

**Fig. 5 Computation study. a** Chemical potential of atoms in Pt nanoparticles on these five carbon supports. **b** Size dependence of diffusion coefficient of Pt nanoparticles on the five support surfaces at 900 °C. **c** Characteristic diffusion length of Pt nanoparticles in two hours versus the contact angles of Pt on these five carbon supports under different temperatures. **d** Measured critical Pt loading versus the theoretically predicted one. The prediction was based on the calculated characteristic diffusion length and measured specific surface area of the carbon supports.

shown in Fig. 5d, the calculated loadings are consistent with the experimental ones well, confirming that maximizing the particle spacing and enhancing MSI can significantly increase the sintering-resistance of supported metal nanocatalysts. We noted that the experimental data for Pt/BP2000 is a bit higher than that predicted by the theory. Such deviation could be ascribed to the confinement effect induced by the narrow microporous channels that connect the internal mesopores[44,45], as revealed by the textural analyses of the carbon supports (Supplementary Figs. 23–25).

**Propane dehydrogenation**. To prove the utility of the critical distance concept, we investigated the sintering resistance of the Pt/C catalysts under realistic conditions for propane dehydrogenation. A reaction temperature of 500 °C, among the highest industrially applied temperatures, was employed to study the catalyst deactivation within laboratory timescales. Two Pt/XC-72R catalysts with Pt loading of 1.0 wt% (less than the critical loading of 3.0 wt% for XC-72) and 5.0 wt% (higher than the critical loading), which exhibit similar particle size but a difference of particle distance (Supplementary Fig. 26), were tested to demonstrate the significant influence of particle distance on the catalyst sintering. Both catalysts exhibited a similar initial $C_3H_8$ conversion of ~20% and the $C_3H_6$ selectivity was around 92% with good replicability in three parallel experiments (Supplementary Fig. 27). During the 500 min continuous operation for propane dehydrogenation at 500 °C, the longer particle distance catalyst (1.0 wt% Pt) exhibited only a little deactivation of $TOF_{C_3H_8}$ (less than 7%) with a low deactivation rate constant of only 0.009 h$^{-1}$ (Fig. 6a). On the contrary, we observed a significant activity decay by 43% for the short particle distance (5.0

wt% Pt) catalyst, corresponding to a higher deactivation rate of 0.089 h$^{-1}$, which is 10 times greater than that of the larger particle distance catalyst. Meanwhile, the sharp contrast of sintering resistance was also found for the Pt/KJ300J catalysts with Pt loading of 1.0 and 8.0 wt% under same propane dehydrogenation (Fig. 6b). Electron microscopy measurements revealed that Pt nanoparticles in the spent low-Pt-loading catalyst retained the uniform particle size and distribution, whereas plenty of large particles of 5–20 nm emerged in the spent high-Pt-loading catalyst (Figs. 6c and d). These results confirmed that the Pt sintering at high temperatures contributed to the fast deactivation of high-Pt-loading catalysts with short particle distance. Note that the sintering of 8.0%Pt/KJ300J catalyst was very limited in H$_2$/Ar at 700 °C for a short duration of 120 min (Fig. 2), while the same catalyst suffered severe at 500 °C for a long duration of 500 min under the condition of realistic propane dehydrogenation reaction (Fig. 6d). Such different sintering phenomena between the sintering test and the realistic propane dehydrogenation reaction could be ascribed to varied sintering time and atmosphere. Particularly, the realistic propane dehydrogenation reaction that involves numerous adsorption and desorption steps towards various carbon-containing intermediates could has a profound influence on the sintering of supported Pt catalysts[46].

**Discussion**

To summarize, we have quantified the critical particle distance for suppressing Pt sintering up to 900 °C on carbon black supports. In situ aberration-corrected HAADF-STEM and computation studies revealed that the particle coalescence sintering was suppressed greatly for the Pt/C catalysts when the particle distance of

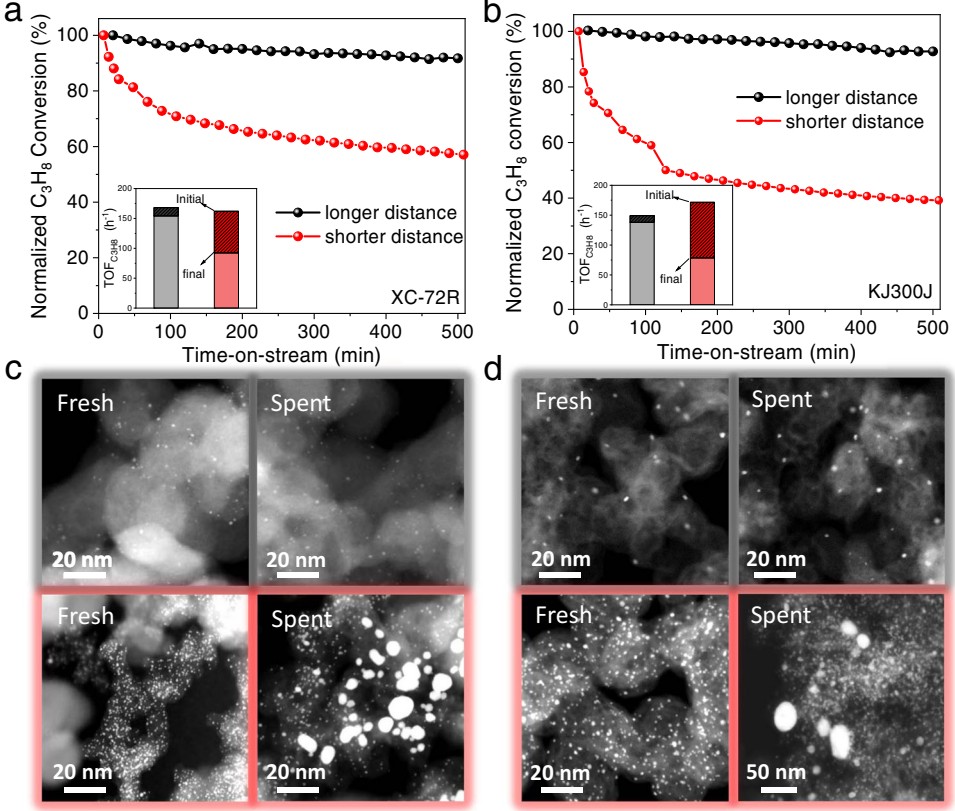

**Fig. 6 Sintering resistance of Pt/C catalysts for propane dehydrogenation. a, b** Normalized $C_3H_8$ conversion versus the time-on-stream on shorter/ longer particle distance catalysts at 500 °C for Pt/XC-72R (**a**) and for Pt/KJ300J (**b**). The insets show the TOF change after reaction. **c, d**, HAADF-STEM images of fresh and spent Pt/XC-72R (**c**) and Pt/KJ300J (**d**).

Pt nanoparticles was above the critical value. The quantified critical particle distance was identified to be dependent on the strength of metal-support interactions. Strengthening the interaction between metal and carbon supports by heteroatom doping could shorten the critical particle distance and accordingly improve the critical metal loading.

It should be pointed out that besides the metal-support interactions the critical particle distance would also vary for different metals and supports and be highly relevant to the particle size and realistic reaction temperature and atmosphere. But our results demonstrated the general validity of enlarging particle distance, which was heavily neglected before, to greatly slow down the catalyst deactivation. Our work would also motivate future works on the synthesis of small sized high-entropy alloy or atomically ordered intermetallic alloy nanoparticle catalysts by high-temperature annealing that is often mandatory to form these two kinds of alloy structures[47,48].

## Methods

**Materials and chemicals.** Commercial carbon black supports, including XC-72R, KJ-300J, KJ-600J, and BP2000 were produced by America Cabot Corporation and Japan Ketjenblack Internation Corporation, respectively. 2,2'-bithiophene (98%) was purchased from J&K Scientific Ltd. All other chemicals were commercially available from Sinopharm Chemical Reagent Co. Ltd., China, including chloroplatinic acid ($H_2PtCl_6·6H_2O$), ruthenium (III) chloride hydrate ($RuCl_3·H_2O$), rhodium (III) chloride hydrate ($RhCl_3·3H_2O$), and iridium (III) chloride hydrate ($IrCl_3·H_2O$). DI water (18.2 MΩ/cm) used in all experiments was prepared by passing through an ultra-pure purification system. All the chemicals were used as received without further purification.

**Synthesis and sintering tests of the Pt/C catalysts.** Pt/C catalysts were prepared by the conventional impregnation method that involved the wet-impregnation of metal salt precursors followed with $H_2$-reduction at 300 °C.

Briefly, a certain amount of $H_2PtCl_6$ was first mixed with 100 mg of carbon black in a 250 mL round-bottom flask containing 100 mL of DI water. After ultrasonic treatment for 2 h, the mixture was subjected to stir overnight before drying by using a rotary evaporator. After being ground in an agate mortar, the resulting dark powder was heated in a tube furnace at 300 °C under flowing 5% $H_2$/Ar for 2 h to get the initial catalysts. Next the sintering experiments were carried out with the fresh catalysts at a commonly used annealing protocols from room temperature to the target temperature of 500, 700, and 900 °C, respectively, under flowing 5% $H_2$/ Ar for 2 h.

The S-BP2000 support was prepared by cobalt-assisted carbonization of molecular precursor, that is, 2,2'-bithiophene, on the surface of BP2000. Briefly, 2.0 g 2,2'-bithiophene and 1.0 g $Co(NO_3)_2·6H_2O$ were dissolved in 120 mL THF. 2.0 g BP2000 was added to form a homogeneous solution by stirring for 4 h. Then the mixture was dried by a rotary evaporator. The dried mixture was subjected to pyrolysis treatment at 800 °C under $N_2$ atmosphere for 2 h with a heating rate of 5 ° C min$^{-1}$. Afterward, cobalt species were removed by leaching in 0.5 M $H_2SO_4$ for 8 h at 90 °C to form the S-BP2000 support. The synthesis and sintering tests of Pt/ S-BP2000 were performed by the same processed as the above Pt/C catalysts.

**In situ aberration-corrected HAADF-STEM measurement.** The in situ heating experiments were obtained on a Wildfire S3 single tilt sample heating system (DENS solutions) inside a probe aberration-corrected JEM ARM200F (JEOL), operated at 200 kV. The Nano-Chip (P.J.H.ST.1) was employed for the experiments and calibrated by DENS solutions. Before the sample loading, holes with diameter of 0.75 μm were milled on SiN membranes with a focused ion beam (Helios Nanolab 600i, FEI). $H_2PtCl_6$-impregnated carbon black was dispersed in n-hexane and then dropped onto the Nano-Chip. A high-angle annular dark field detector provided an incoherent projection image of the specimen with a signal intensity proportional to the amount of material and $Z^2$ (Z, atomic number), which is also known as Z-contrast. The in situ aberration-corrected HAADF-STEM measurement was carried out with the temperatures-time program shown in Fig. 4c.

**Catalytic propane dehydrogenation.** The catalytic propane dehydrogenation reaction was performed in a fixed-bed quartz tube reactor and the products were analyzed by an online GC-14C gas chromatograph equipped with a flame ionization detector and a thermal conductivity detector. Before the dehydrogenation reaction, the catalysts were reduced in $H_2$/Ar at 200 °C for 1 h. After reduction, the

reactor was heated to 500 °C as operating temperature and fed with the gas mixture for 30 min. Typically, the reactant gas mixture was composed of 10 vol % propane with a balance of $N_2$ ($N_2:H_2:C_3H_8$ = 8:1:1, vol.) from a pressurized gas-mixture cylinder (total flow rate of 15 mL min$^{-1}$). The weight hourly space velocity (WHSV) of propane was controlled to be around 2 h$^{-1}$, respectively. After pelleting the catalysts and quartz sand to 40-60 mesh to control the same bed height, the same dosage of Pt (1.0 mg) was used to ensure the similar initial conversion. The Turnover frequency (TOF) was calculated as moles of $C_3H_8$ conversion per mole of Pt per hour.

$$\text{TOF}_{C_3H_8} = \frac{F(C_3H_8) \times 60 \div 22.4 \times X_{(C_3H_8)}}{m_{cat} w_{Pt} \div 195.1} \quad (9)$$

where $F$ ($C_3H_8$) represents the flow rate of propane, $X_{(C_3H_8)}$ and $m_{cat}$ are the conversion of propane and catalyst loading, respectively, and $W_{Pt}$ is the percentage of Pt weight loading in the catalyst.

The deactivation rate constant (h$^{-1}$), which is derived from the first order deactivation model to the evaluate catalyst stability:

$$K_d = \frac{\ln \frac{1-X_{final}}{X_{final}} - \ln \frac{1-X_{innitial}}{X_{innitial}}}{t} \quad (10)$$

where $X_{innitial}$ and $X_{final}$, respectively, represent the conversion measured at the initial and final period of an experiment, and $t$ represents the reaction time (h), $K_d$ is the deactivation rate constant (h$^{-1}$). High $K_d$ value means rapid deactivation, that is, low stability.

**Characterization**. HAADF-STEM images were obtained on FEI Talos F200X operated at 200 kV. XRD analyses were carried with a Japan Rigaku DMax-γA rotation anode X-ray diffractometer equipped with graphite monochromatized Cu-K radiation (λ = 1.54178 Å). $N_2$ sorption analysis was conducted using an ASAP 2020 accelerated surface area and porosimetry instrument (Micromeritics), equipped with automated surface area, at 77 K using BET calculations for the surface area. The pore size distribution plot was analyzed from the adsorption branch of the isotherm based on the quenched-solid density functional theory (QSDFT). Raman scattering spectra were recorded with a Renishaw System 2000 spectrometer using the 514.5 nm line of Ar+ for excitation.

## Data availability

All data presented in this study are available from the corresponding authors upon request.

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

## Acknowledgements

This work was supported by the National Key Research and Development Program of China (Grant 2018YFA0702001, 2018YFA0208603, and 2019YFA0307900), the National Natural Science Foundation of China (Grant 21671184, 11874334, 21903077, 91945302, and 22072118), the Fundamental Research Funds for the Central Universities (Grant WK2060190103), Youth Innovation Promotion Association CAS (2020458), Chinese Academy of Sciences (QYZDJ-SSW-SLH054) and the Joint Funds from Hefei National Synchrotron Radiation Laboratory (Grant KY2060000107 and KY2340000115), and the Recruitment Program of Thousand Youth Talents.

## Author contributions

H.-W.L. and P.Y. conceived and designed the project. P.Y. synthesized the catalysts and performed the sintering tests. P.Y. and L.-L.Z carried out the catalysts characteristics. Y.L. performed the in situ aberration-corrected HAADF-STEM studies. W.-X.L. and S.-L.H. performed the calculation. P.Y. and Z.W. performed the catalytic propane dehydrogenation under the direction of K.Q. and W.H. P.Y., S.-L. H., H.-W.L., and W.-X.L. co-wrote the manuscript. H.-F.X. supplied the helpful discussions for the manuscript. All authors discussed the results and commented on the manuscript.

## Competing interests

The authors declare no competing interests.
