## [Peer Review File · Nature Communications]

REVIEWER COMMENTS

Reviewer #1 (Remarks to the Author):

This is a nice paper to collect and analyze data on particle distributions and sintering (collision?) with very simplified model. Probably, such discussions can be possible only under very unified conditions. This work would give some hints on how to expect sintering under limited conditions. However, it cannot be general ways. This work is based on mostly geometric factors but important physico-chemical factors are not included.

- 1) Sintering would significantly depend on surface energy problems.
- 2) Mobility of particle on supports would significantly depend on energies between surface and particle
- 3) The above factors have to be considered thermodynamically.
- 4) More importantly, kinetic factors have high temperature dependence.

I do not think that this work well include necessary physico-chemical backgrounds and cannot be generalized as general science. Therefore, I do not have high impact enough to get publication in Nature Communications.

Reviewer #2 (Remarks to the Author):

Liang and co-workers have studied the impact of metal loading and support specific surface area (SA) on the stability of Pt/C catalysts for sintering in H₂ and in propane/H₂. The intention of this study to arrive at a safe interparticle distance between metal nanoparticles in catalysis to mitigate sintering. This study is of high quality and addresses a feature that is often overlooked in the study of catalysis, i.e. nanoparticle spacing. In situ TEM studies are also presented that provide further insights. However, a number of issues need serious attention before publication in Nature Communications is feasible.

1. A major experimental issue for Pt/C sintering studies in H₂/Ar at elevated temperatures seems to me the possible gasification of the support. Carbon can react with H₂ to form CH₄ which is probably catalysed by Pt. Please measure Pt loading of Pt/C samples before and after high temperature treatments to establish whether or not the gasification of the support takes place. If so then study also the specific surface area and micropore volumes of the samples before and after (see also point 2).
2. Although the authors address the possible role of micropores to stabilize Pt nanoparticles this issue should be studied more in depth. Support BP-2000 displays more micropores that seems to lead to higher stability than support KJ600J with similar SA (see results in Fig. 3b). I propose that for all supports the authors use their physisorption data to report the micropore volumes, mesoporous specific surface areas (from the T-plot), the total pore volume at p/p₀=0.95 and the pore size distribution. In this way they can provide a more complete textural analysis which might lead to other correlations of stability

then only with interparticle distances. I look forward to these results.

3. In my view Figure 1 is not correct for the carbon black supports that the authors have used. The primary carbon particles are not isolated from each other but are present as aggregates. In the contact points between the primary carbon particles, Pt particles may be preferably located (due to capillary forces during drying after impregnation). Therefore, at these contact points locally higher densities of Pt nanoparticles may be present that lead to sintering rates above average. This complexity of carbon black should be considered in the paper.

4. For sample 8wt% Pt/KJ300J sintering in H₂/Ar did not take place at 700C (see Fig. 2-b) whereas extensive sintering took place for propane dehydrogenation at 550C (see results in Fig. 5). The authors do not discuss this big difference between these results that are not easy to reconcile. Please consider and discuss.

5. Sulfur doping is suggested to lead to SMSI that reduces sintering. This is difficult to understand for me. What is the sulfur content of the Pt/C samples before and after heat treatment in H₂/Ar at elevated temperatures? I would expect that the S has been removed as H₂S and the sulfur levels will be low.

6. The propane dehydrogenation experiments need more detail. What grain sizes of the catalyst have been used (powders lead to irreproducible results). What conversion levels of propane were obtained? Report primary data in SI next to the normalised data in the paper.

In summary, this paper warrants publication in Nature Communications but some important issues summarised above need to be addressed first.

Reviewer #3 (Remarks to the Author):

The work from Yin et al. reported the quantification of the “safe” inter-particle distance using the model system of Pt/carbon. Mostly based on microscopic evidences, the authors correlated the stabilization and transformation of the supported Pt nanoparticles with the inter-particle distances, which are further optimized for propane dehydrogenation. However, the essence of the “safe” inter-particle distance is the loading density, and the authors did not probe the metal-metal and metal-support interactions that fundamentally determines particle sintering. Therefore, more work is anticipated to reveal further insights that ultimately contribute to the rational design of anti-sintering catalysts.

1. The evidences for the sintering of the supported Pt nanoparticles are mainly based on the microscopic images and the XRD patterns, both of which might introduce errors for precise quantification. The three-dimensional architecture makes the size identification not easy, as the images are two-dimensional. Therefore, in the case of two particles overlapping with each other from the Z direction which is highly likely for high loading densities, or possessing anisotropic morphologies, things can get tricky. On the other hand, the full-width at half maximum in the Debye-Scherrer equation indicates the size of the domain, which is not the same as the size of the particles for polycrystalline structures as evident in Fig. 4 and S6. Therefore, while the conclusion that low-loading leads to better sintering resistance capabilities is correct according to empirical evidences, the authors should be careful analyzing the data.

Another route to precisely quantifying the size of the particles, or the number of the active sites is chemisorption, which will greatly complement the microscopic and XRD results.

2. The authors mentioned that BP2000 has a higher resistance towards particle growth is due to the enrichment of the micropores. Since the size of the micropores (less than 2 nm) is smaller than the size of the nanoparticles (3 nm), can the authors elaborate on this? Pore-confined migration and stabilization of active species have drawn continuous attentions during the past few years.

3. For the enhanced metal-support interactions between Pt and S-BP2000, XPS or EELS would be desired to demonstrate the formation of the interfacial S-Pt bonds.

4. The authors used in-situ high-resolution STEM to discuss the stages where particle coalescence and Ostwald ripening occur. Since they did not specify the in-situ atmosphere during this process, I would assume they used vacuum when heating and quenching the sample. In that case, the collected results could be very different from the actual case in 5% hydrogen balanced with argon, which makes the conclusion doubtful. In addition, the formation, migration and absence of the atomic Pt species in Fig. 4b could be caused by the “knock-on” damage given the condition of 200 kV with atomic resolutions. Therefore, it would be nice if the authors could provide time-resolved STEM images to eliminate this possibility, which is important to evaluate the validity of the conclusion.

5. Again for Fig. 4d and 4e, the authors used the two-dimensional area to represent the three-dimensional particle size, which only works for isotropic particles and not the case here.

6. For the dehydrogenation measurement, can the authors also provide the activity based on the active sites of Pt, like TOF, for both samples, instead of percentage conversion? That will present more useful information for comparison.

Reviewer #1 (Remarks to the Author):

This is a nice paper to collect and analyze data on particle distributions and sintering (collision?) with very simplified model. Probably, such discussions can be possible only under very unified conditions. This work would give some hints on how to expect sintering under limited conditions. However, it cannot be general ways. This work is based on mostly geometric factors but important physico-chemical factors are not included.

Response: We really appreciate the reviewer's professional comments that push us to greatly promote the quality of the work. We agree with the reviewer that the sintering is a complex phenomenon, which is dependent on physical-chemical factors. Accordingly, we systematically studied the physical-chemical fundamentals behind the sintering of Pt/C from the thermodynamics to kinetics. In particular, we experimentally measured the average contact angle of Pt nanoparticles on carbon supports and accordingly calculated the surface energy and chemical potential of Pt nanoparticles for the practical Pt/C catalyst systems. Further, we calculated the diffusion coefficient and diffusion length of Pt nanoparticles on carbon surface based on the chemical potential and the measured contact angle. In consequence, we theoretically predict the critical loading of Pt nanoparticles on different carbon supports. We believe that these complementary results strongly evidence the generalization of this work: maximizing the particle spacing and enhancing the metal-support interaction can in large extent increase the sintering-resistance of supported metal catalysts.

- 1) Sintering would significantly depend on surface energy problems.
- 2) Mobility of particle on supports would significantly depend on energies between surface and particle.

Response: We agree with the reviewer that the sintering is highly dependent on the surface energy of metal nanoparticles as well as the energies between surface and particle (that is, the interfacial adhesion energy). The reviewer's comments inspired us

to consider these physical-chemical factors for better understanding the experimentally observed particle distance-dependent sintering behavior of the Pt/C catalysts.

To reveal the underlying physics of the particle distance dependent sintering of the Pt/C catalysts, we performed the systematical computational studies. According to the above in situ aberration-corrected HAADF-STEM studies, the Pt sintering is found to proceed majorly via the PMC path, which depends not only on the particle spatial and size distribution but also the metal-support interaction (MSI). Corresponding rate is determined essentially by particle diffusion coefficient on supports as derived in our previous work¹:

$$D_p(R) = \frac{K_\alpha}{R^4} \exp\left[\frac{\Delta\mu(R)}{k_B T}\right] \exp\left[-\frac{E_{\text{act}}^m - S^m E_{\text{adh}}}{k_B T}\right] \quad (1)$$

where R is the radius of curvature of the supported particle, K_α is the structure factor related to the contact angle α of metal particle with the support, k_B is Boltzmann's constant, T is temperature. E_{act}^m is the self-activation energy of metal atoms that characterizes its formation and migration on the surface of nanoparticle and is considered as one-third of the cohesive energy of bulk metal (1.93 eV for Pt)². S^m is the area of the Pt atom (6.17 \AA^2), and E_{adh} is the interfacial adhesion energy between metal and support and is closely related to α according to the Young-Dupre' equation²

$$E_{\text{adh}} = -\gamma_m(1 + \cos\alpha) \quad (2)$$

where γ_m is the experimental surface energy of the Pt bulk (0.155 eV/\AA^2). $\Delta\mu(R)$ is the chemical potential of metal atoms. $\Delta\mu(R)$ in supported nanoparticles with respect to the bulk counterpart can be calculated approximated by Gibbs-Thomson (G-T) relation³

$$\Delta\mu(R) = \frac{2\Omega\gamma_m}{R} \quad (3)$$

where Ω is the volume of Pt atom (15.33 \AA^3). For small nanoparticles, γ_m becomes size dependent due to the increase of low coordination sites, and could be calculated by:

$$\gamma_m(R) = \frac{\bar{C}^m(R) - 12 E_c}{2 \times 12 S^m} \quad (4)$$

where $\bar{C}^m(R)$ is the weighted surface coordination number for the Wulff structure⁴, S^m is the metal atom surface area and E_c is the cohesive energy of Pt. Accordingly, the size-dependent surface energy was confirmed and showed a convergence to the surface energy of bulk Pt (Fig. R1a).

The structure factor related with the contact angle α between the metal particle and support K_α could be calculated by Equation 5.

$$K_\alpha = \frac{3\nu_p C_\alpha \Omega^2}{2\pi\alpha_1} \quad (5)$$

$$C_\alpha = \begin{cases} \sin\alpha, & \alpha < \pi/2 \\ 1, & \alpha \geq \pi/2 \end{cases} \quad (6)$$

where ν_p is the vibrational frequency of metal atom on support surface, R is the radius of curvature of nanoparticle on the support.

When a metal particle with a given volume V_0 lands on the support of interest, it tends to adopt into a truncated sphere shape with α , and the corresponding R is determined by

$$R = \sqrt[3]{3V_0/4\pi\alpha_1} \quad (7)$$

where

$$\alpha_1 = (2 - 3\cos(\alpha) + \cos^3(\alpha))/4 \quad (8)$$

We performed the high resolution transmission electron microscopy (HRTEM) observations to roughly extract the corresponding α of Pt nanoparticles on carbon supports and (Fig. R2). By statistically analyzing the HRTEM images, the average α was derived for each sample and plotted in Fig. R3. It was found that XC-72R has the largest α of 125.4° representing a relatively weak MSI, followed by KJ300J (124.1°), KJ600J (118.9°), BP2000 (116.2°), and S-BP2000 (108.7°) representing a relatively stronger MSI. Corresponding E_{adh} between Pt and five carbon supports based on the α data were calculated. Based on the extracted α and taking into account of the size effect on γ_m , $\Delta\mu$ of Pt atoms on these carbon supports was calculated and plotted with respect to the particle in Fig. R1b. At a given particle size, the Pt nanoparticles on XC-72R was found to have a highest $\Delta\mu$, whereas have a lowest one on S-BP2000 due to the strengthened MSI. We then calculated D_p of Pt nanoparticles versus the particle size on different carbon supports at 900 °C, as shown in Fig. 5b. For small 2 nm Pt of interest, the calculated D_p is as high as 24.3 nm² s⁻¹ on XC-72R, but only 1.6 nm² s⁻¹ on S-BP2000. Such difference by a factor of fifteen along with the change in α of 20° is significant, indicating the great influence of MSI on nanoparticle diffusion. Note that D_p decreases dramatically with the particle size, due to the inverse fourth power relationship on R (Eq. 1).

For a time period of τ , the spatial dimension of the Pt nanoparticle diffusion via Brownian motion can be measured by a characteristic length L :

$$L=[D_p(\alpha) \tau]^{1/2} \quad (9)$$

As shown in Fig. R4, for $\tau = 2$ hours, L is insensitive to α at a relatively low temperature of 700 °C. When increasing the temperature up to 900 °C, L increases considerably and becomes sensitive to α , and specifically, 201 nm on XC-72R (125.4°) versus 52 nm on S-BP2000 (108.7°). Small L on S-BP2000 than that on XC-72 R implies low probability for two particles to collide and merge into a larger particle, corroborating well with the trend behavior of experimentally extracted critical particle distance. Supposing the support with an area of $\pi(L/2)^2$ only supports one particle, the critical loading of the supported Pt nanoparticles with a volume of $4\pi\alpha_1 R^3/3$ can be estimated by:

$$\text{Loading} = 16S_m R^3 \alpha_1 m_{Pt} / (3L^2 \Omega), \quad (10)$$

for a given mass specific surface area S_m of the support, where m_{Pt} is Pt atom mass (3.24×10^{-22} g/atom)². For 2 nm Pt dispersed on five carbon supports and aging at 900 °C for 2 hours, the maximum loadings are calculated and compared with the experiments. As shown in Fig. R5, the calculated loadings are consistent with the experimental ones well, confirming that maximizing the particle spacing and enhancing MSI can significantly increase the sintering-resistance of supported metal nanocatalysts. We noted that the experimental data for Pt/BP2000 is a bit higher than that predicted by the theory. Such deviation could be ascribed to the confinement effect induced by the narrow microporous channels that connect the internal mesopores^{5, 6}, as revealed by the textural analyses of the carbon supports (Supplementary Figs. 23-25, please see our response to Reviewer #3 for more details on this issue).

Figure R1. Size/support-dependent chemical potential. (a) Size dependence of surface energy of Pt nanoparticles based on the constructed spherical (circles) and Wulff (squares) structures with a convergence to the bulk Pt surface energy (green dashed line). (b) Size dependence of chemical potential of Pt atoms on different carbon supports.

Figure R2. Measured contact angle of Pt nanoparticles on the five carbon supports by HRTEM.

Figure R3. Average contact angle of Pt nanoparticles on the five carbon supports.

Figure R4. Size-, support- and temperature-dependent mobility of Pt nanoparticles. (a) Size dependence of the diffusion coefficient of Pt nanoparticles on the five carbon supports. (b) Diffusion length of Pt nanoparticles in two hours versus the contact angles of Pt on these five carbon support surfaces under different temperatures.

Figure R5. Measured critical Pt loading versus the theoretically predicted loading. The prediction is based on the calculated diffusion length (Eq. 10) and measured specific surface area of the carbon supports.

The above experimental and computational results and related discussion have been added in the revised manuscript (Page 9 and Fig. 5) and Supplementary Information (Figs. S20-S22)

3) The above factors have to be considered thermodynamically.

Response: As responded above, we have thermodynamically considered the influence of the particle size and interfacial adhesion energy on the surface energy and chemical potential of carbon supported Pt nanoparticles (Fig. R1). After identifying these thermodynamic parameters, we could further calculate the particle's diffusion coefficient (Fig. R4a) and the diffusion length at a given temperature and time duration (Fig. R4b). Eventually, we even predicted the critical loading of Pt nanoparticles on different carbon supports (Fig. R5).

4) More importantly, kinetic factors have high temperature dependence.

Response: We have also considered the kinetic factors that have high temperature dependence, including the particle's diffusion coefficient (Fig. R4a) and the diffusion length at a given temperature and time duration (Fig. R4b). Kinetically, the diffusion or mobility of nanoparticles on the support surface is highly temperature dependent, because atomically the thermal energy ($k_B T$) of the system (k_B is Boltzmann's constant, T is temperature) provides the force driving the atom's migration or flow. As shown in Fig. R4b, the diffusion lengths of Pt nanoparticles on these five support surfaces show a significant temperature dependence. For example, on XC-72R support surface, the calculated diffusion length of Pt nanoparticles increases from 45 nm to 215 nm in two hours as the temperature increases from 700 °C to 900 °C.

I do not think that this work well include necessary physico-chemical backgrounds and cannot be generalized as general science. Therefore, I do not have high impact enough to get publication in Nature Communications.

Response: As we responded, the above complementary experimental and computational data on physical-chemical fundamentals has involved all the mentioned issues, including surface energy, energies between supports and particle, sintering

thermodynamic and kinetic factors. These results have fully showed the generalization of this work: maximizing the particle spacing and enhancing the metal-support interaction can in large extent increase the sintering-resistance of supported metal catalysts.

We agree with the review that the quantified critical particle distance is not a general value. Besides the metal-support interactions the critical particle distance would also vary for different metals and supports and be highly relevant to the realistic reaction temperature and atmosphere. Therefore we have deleted the words of “general” and “30 nm” from the Abstract, Introduction, and Discussion parts in the revised manuscript.

Reviewer #2 (Remarks to the Author):

Liang and co-workers have studied the impact of metal loading and support specific surface area (SA) on the stability of Pt/C catalysts for sintering in H₂ and in propane/H₂. The intention of this study to arrive at a safe interparticle distance between metal nanoparticles in catalysis to mitigate sintering. This study is of high quality and addresses a feature that is often overlooked in the study of catalysis, i.e. nanoparticle spacing. In situ TEM studies are also presented that provide further insights. However, a number of issues need serious attention before publication in Nature Communications is feasible.

1. A major experimental issue for Pt/C sintering studies in H₂/Ar at elevated temperatures seems to me the possible gasification of the support. Carbon can react with H₂ to form CH₄ which is probably catalysed by Pt. Please measure Pt loading of Pt/C samples before and after high temperature treatments to establish whether or not the gasification of the support takes place. If so then study also the specific surface area and micropore volumes of the samples before and after (see also point 2).

Response: Many thanks for the reviewer’s comments on this issue. As suggested by

the reviewer, we further measured Pt loading, the specific surface area and micropore volumes of each Pt/C sample after thermal treatments at different temperature of 500, 700, and 900 °C to assess the possible gasification of the carbon supports with the presence of Pt. As shown in Fig. R6, the Pt loading for all the four carbon black supported catalysts maintained unchanged upon the thermal treatments (3%, 8%, 15%, and 25% Pt for XC-72, KJ300J, KJ600J, and BP2000, respectively). Besides, the N₂ sorption tests showed no obvious change of these Pt/C catalysts in specific surface area and micropore volumes (Fig. R7). These results suggested the negligible gasification of these commercial carbon black support at high temperatures up to 900 °C even when loading with Pt nanoparticles.

Figure R6. Pt loading of the four carbon black supported catalysts upon thermal treatments at 500, 700, and 900 °C.

Figure R7. Isothermal physisorption curves, the corresponding micropore volumes and specific surface area of the carbon black supported catalysts upon thermal treatments at 500, 700, and 900 °C.

The above results and discussion have been added in the revised manuscript (Page 4) and Supplementary Information (Figs. S2-S3)

2. Although the authors address the possible role of micropores to stabilize Pt nanoparticles this issue should be studied more in depth. Support BP-2000 displays more micropores that seems to lead to higher stability than support KJ600J with similar SA (see results in Fig. 3b). I propose that for all supports the authors use their physisorption data to report the micropore volumes, mesoporous specific surface areas (from the T-plot), the total pore volume at $p/p_0=0.95$ and the pore size distribution. In this way they can provide a more complete textural analysis which might lead to other correlations of stability then only with interparticle distances. I look forward to these results.

Response: Many thanks for the reviewer's comments on this issue. As suggested by the reviewer, we further supplemented the micropore volumes, mesoporous specific surface areas, the total pore volume at $p/p_0=0.95$ and the pore size distribution of the

carbon black supports (Fig. R8). Clearly, XC-72R is a kind of solid carbons with little micropore and mesopore, while KJ300J, KJ600J, and BP2000 are hierarchically porous carbon with high specific surface areas. In particular, BP2000 has a much higher ratio of micropore volume and micropore surface area than KJ300J and KJ600J.

Before considering the possible role of micropores of BP2000, as we respond to Reviewer #1, we have additionally studied the physical-chemical fundamentals behind the sintering of Pt/C from the thermodynamics to kinetics. We experimentally measured the average contact angle of Pt nanoparticles on carbon supports and accordingly calculated the surface energy and chemical potential of Pt nanoparticles for the practical Pt/C catalyst systems. We then calculated the diffusion coefficient and length of Pt nanoparticles on carbon surface based on the chemical potential and the measured contact angle. Finally, we could predict the critical loading of Pt nanoparticles on different carbon supports, which is consistent well with the experimentally measured critical loading (Fig. R5). These results confirmed that maximizing the particle spacing and enhancing the metal-support interaction could in large extent increase the sintering-resistance of supported metal catalysts.

We noted some deviations of the experimentally measured critical Pt loading from the predicted one, particularly for the case of Pt/BP2000 (Fig. R5). Such deviation could be ascribed to the confinement effect induced by the numerous pores in BP2000. We observed narrow channels of ~ 2 nm that connect the internal carbon mesopores in the carbon supports, which means that Pt particles located in interior mesopores are not easy to agglomerate through the PMC mechanism owing to the obstruction of the narrow micropore channels at the outlet (please see our response to Reviewer #3 for more details on this issue).

Figure R8. Textural analyses of the four carbon black supports, including the pore size distribution (a), mesoporous specific surface areas (b), the total pore volume at $p/p_0=0.95$ (c), and the micropore volumes (d).

The above results and discussion have been added in the revised manuscript (Page 11) and Supplementary Information (Fig. S23)

3. In my view Figure 1 is not correct for the carbon black supports that the authors have used. The primary carbon particles are not isolated from each other but are present as aggregates. In the contact points between the primary carbon particles, Pt particles may be preferably located (due to capillary forces during drying after impregnation). Therefore, at these contact points locally higher densities of Pt nanoparticles may be present that lead to sintering rates above average. This complexity of carbon black should be considered in the paper.

Response: According to the reviewer's comments, we performed additional TEM measurements to analyze the homogeneity of the Pt nanoparticle distribution on the carbon support. The TEM images showed the highly homogeneous distribution of Pt

nanoparticles on all the four carbon black supports, and we did not observed the enrichment of Pt nanoparticles at the carbon particle contact region (Fig. R9). We agree with the reviewer that the primary carbon particles are not isolated from each other but are present as aggregates. Figure 1 is just a simplified model for demonstrating the effect of inter-particle distance in sintering, without considering the relatively complex carbon structure.

Figure R9. TEM images of different Pt/C catalysts, showing the uniform Pt nanoparticle distribution throughout the carbon black supports.

4. For sample 8wt% Pt/KJ300J sintering in H₂/Ar did not take place at 700C (see Fig. 2-b) whereas extensive sintering took place for propane dehydrogenation at 500C (see results in Fig. 5). The authors do not discuss this big difference between these results that are not easy to reconcile. Please consider and discuss.

Response: Many thanks for the reviewer's comments on this issue. The different sintering phenomena between the sintering test and the realistic propane dehydrogenation reaction could be ascribed to varied sintering time and atmosphere. For the duration time was 120 min for the sintering test at 700 °C, while the duration

time of propane dehydrogenation reaction at 500 °C was 500 min. The long-term thermal treatments would induce sintering even at a lower temperature. Besides, the realistic propane dehydrogenation reaction that involves numerous adsorption and desorption steps towards various reaction intermediates could have a profound influence on the sintering behavior of supported Pt catalysts⁷.

5. Sulfur doping is suggested to lead to SMSI that reduces sintering. This is difficult to understand for me. What is the sulfur content of the Pt/C samples before and after heat treatment in H₂/Ar at elevated temperatures? I would expect that the S has been removed as H₂S and the sulfur levels will be low.

Response: Many thanks for the reviewer's comments on this issue. As suggested by the reviewer, we measured the sulfur content by XPS for the Pt/S-BP sample before and after heat treatment in H₂/Ar at 900 °C. As the reviewer expected, the sulfur content of the Pt/S-BP2000 decreased gradually from 3.20 at% for the initial S-BP2000 to 1.02 at% after the thermal treatment (Fig. R10). We further investigated the change of sulfur doping upon the thermal treatment by energy dispersive spectroscopy (EDS) elemental mapping. Interestingly, we observed the spatial overlapping of Pt and residual sulfur elements (Fig. R11), which strongly suggested the key role of the doped sulfur atoms as anchoring sites for Pt nanoparticles even at high temperature^{8,9}.

Figure R10. XPS survey spectra of the S-BP2000 support and the Pt/S-BP2000 catalyst after annealing treatment at 900 °C.

Figure R11. EDS mapping of Pt/S-BP2000 after the treatment at 900 °C, showing the spatial overlapping of Pt and S elements.

The above results and discussion have been added in the revised manuscript (Page 6) and Supplementary Information (Fig. S7-S8)

6. The propane dehydrogenation experiments need more detail. What grain sizes of the catalyst have been used (powders lead to irreproducible results). What conversion levels of propane were obtained? Report primary data in SI next to the normalised data in the paper.

Response: Many thanks for the reviewer's comments on this issue. After pelleting the catalysts and quartz sand to 40-60 mesh to control the same bed height, the same dosage of Pt (1.0 mg) was used to ensure the similar initial conversion. The catalytic propane dehydrogenation reaction was performed in a fixed-bed quartz tube reactor and the products were analyzed by an online GC-14C gas chromatograph equipped with a flame ionization detector and a thermal conductivity detector. Before the dehydrogenation reaction, the catalysts were reduced in H₂/Ar at 200 °C for 1 h. After reduction, the reactor was heated to 500 °C as operating temperature and fed with the gas mixture for 30 minutes. Typically, the reactant gas mixture was composed of 10 vol % propane with a balance of N₂ (N₂: H₂: C₃H₈=8:1:1, vol.) from a pressurized gas-mixture cylinder (total flow rate of 15 mL·min⁻¹). The weight hourly space velocity (WHSV) of propane

was controlled to be around 2 h^{-1} . We repeated three parallel experiments to verify the replicability of results (Figure R12). The initial C_3H_8 conversion was at a same level ($\sim 20\%$) and the C_3H_6 selectivity was around 92%. These results all suggested the significant influence of inter-particle distance on the catalyst stability.

Figure R12. Propane dehydrogenation for Pt/XC-72R and Pt/KJ300 at 500 °C

The above results and discussion have been added in the revised manuscript (Page 11) and Supplementary Information (Fig. S27)

Reviewer #3 (Remarks to the Author):

The work from Yin et al. reported the quantification of the “safe” inter-particle distance using the model system of Pt/carbon. Mostly based on microscopic evidences, the authors correlated the stabilization and transformation of the supported Pt nanoparticles with the inter-particle distances, which are further optimized for propane dehydrogenation. However, the essence of the “safe” inter-particle distance is the loading density, and the authors did not probe the metal-metal and metal-support interactions that fundamentally determines particle sintering. Therefore, more work is anticipated to reveal further insights that ultimately contribute to the rational design of

anti-sintering catalysts.

Response: Many thanks for the reviewer's comments here. According to the comments from Reviewers #1 and #3, we have additionally studied the physical-chemical fundamentals behind the sintering of Pt/C from the thermodynamics to kinetics. We experimentally measured the average contact angle of Pt nanoparticles on carbon supports and accordingly calculated the surface energy and chemical potential of Pt nanoparticles for the practical Pt/C catalyst systems. We then calculated the diffusion coefficient and length of Pt nanoparticles on carbon surface based on the chemical potential and the measured contact angle. Finally, we could predict the critical loading of Pt nanoparticles on different carbon supports, which is consistent well with the experimentally measured critical loading (Fig. R5). These experimental and computational data on physical-chemical fundamentals fully demonstrated the generalization of this work: maximizing the particle spacing and enhancing the metal-support interaction can in large extent increase the sintering-resistance of supported metal catalysts.

1. The evidences for the sintering of the supported Pt nanoparticles are mainly based on the microscopic images and the XRD patterns, both of which might introduce errors for precise quantification. The three-dimensional architecture makes the size identification not easy, as the images are two-dimensional. Therefore, in the case of two particles overlapping with each other from the Z direction which is highly likely for high loading densities, or possessing anisotropic morphologies, things can get tricky. On the other hand, the full-width at half maximum in the Debye-Scherrer equation indicates the size of the domain, which is not the same as the size of the particles for polycrystalline structures as evident in Fig. 4 and S6. Therefore, while the conclusion that low-loading leads to better sintering resistance capabilities is correct according to empirical evidences, the authors should be careful analyzing the data. Another route to precisely quantifying the size of the particles, or the number of the active sites is chemisorption, which will greatly complement the microscopic and XRD results.

Response: Many thanks for the reviewer's comments on this issue. As suggested by the reviewer, the microscopic and XRD results may introduce errors for precise quantification and the quantification of the active sites will greatly complement above shortcomings. Here, we additionally measured the electrochemically active surface area (ECSA) for the Pt/C catalysts by the electrochemical CO-stripping technique, which is the reliable approach to assess the surface area of Pt/C catalysts via electrochemical oxidation of an adsorbed CO monolayer over Pt surface¹⁰. With the treatment temperature increased, the ECSA of different Pt/C system was gradually decreased and remained at higher levels equivalent to about 3 nm for the 900 °C sample (Fig. R13). Once exceeding the upper Pt loading on each carbon support, we observed a significantly reduced ECSA for all the samples at 900 °C, indicating severe sintering and greatly losing of Pt active sites. The CO-stripping results were well consistent with microscopic and XRD data; all the data together strongly evidenced the inter-particle distance dependent behavior.

Figure R13. CO-stripping of different Pt/C catalysts and the corresponding ECSA.

The above results and discussion have been added in the revised manuscript (Page 5) and Supplementary Information (Fig. S6)

2. The authors mentioned that BP2000 has a higher resistance towards particle growth is due to the enrichment of the micropores. Since the size of the micropores (less than 2 nm) is smaller than the size of the nanoparticles (3 nm), can the authors elaborate on this? Pore-confined migration and stabilization of active species have drawn continuous attentions during the past few years.

Response: Many thanks for the reviewer's comments on this issue. As previous work reported^{5, 6}, narrow channels of ~2 nm were observed connecting the internal carbon mesopores in carbon black supports, which means that Pt particles located in interior mesopores are not easy to agglomerate through the PMC mechanism owing to the obstruction of the narrow micropore channels at the outlet (Fig. R14). Despite the absence of larger, mesopore-like openings, the Pt atoms of particles in the interior of the carbon supports may remain easily mobile through microporous channels via OR mechanisms. In this case, the abundance of micropores and complex pore structures increases the difficulty of mass transfer among particles via PMC/OR mechanisms. On the other hand, we compared the changes of micropore volume before and after loading of Pt on KJ600 and BP2000 (Fig. R15) and found that the micropore volume on Pt/KJ600 did not change much after loading, but the micropore volume of Pt/BP2000 obviously decreased. This result revealed that some micropores in BP2000 were occupied by Pt nanoparticles, which demonstrated some micropores were used to restrict small particles (< 2 nm) sintering on BP2000.

Figure R14. Schematic illustration of complex micropores and mesopores structure.

Figure R15. Micropore volume changes before and after loading on KJ600 and BP2000.

The above results and discussion have been added in the revised manuscript (Page 11) and Supplementary Information (Fig. S24-S25)

3. For the enhanced metal-support interactions between Pt and S-BP2000, XPS or EELS would be desired to demonstrate the formation of the interfacial S-Pt bonds.

Response: Many thanks for the reviewer's comments on this issue. As suggested by the reviewer, we performed XPS and energy dispersive spectroscopy (EDS) elemental mapping analyses to demonstrate the formation of the interfacial Pt-S bonds. We observed a significant 0.3 eV shift of the Pt 4f peak to a lower bind energy for Pt/S-BP2000 compared to Pt/BP2000 (Fig. R16), confirming the electron donation from the S to Pt by the interfacial Pt-S bonds¹¹. Meanwhile, the spatial overlapping of Pt and S

elements in EDS mapping further suggested the formation of interfacial Pt-S bonds even at high temperatures (Fig. R11, see our response to Reviewer #2 for more details of EDS data).

Figure R16. XPS spectra of Pt 4f on Pt/S-BP2000 and Pt/BP2000.

The above results and discussion have been added in the revised manuscript (Page 6) and Supplementary Information (Fig. S9)

4. The authors used in-situ high-resolution STEM to discuss the stages where particle coalescence and Ostwald ripening occur. Since they did not specify the in-situ atmosphere during this process, I would assume they used vacuum when heating and quenching the sample. In that case, the collected results could be very different from the actual case in 5% hydrogen balanced with argon, which makes the conclusion doubtful. In addition, the formation, migration and absence of the atomic Pt species in Fig. 4b could be caused by the “knock-on” damage given the condition of 200 kV with atomic resolutions. Therefore, it would be nice if the authors could provide time-resolved STEM images to eliminate this possibility, which is important to evaluate the validity of the conclusion.

Response: Many thanks for the reviewer’s comments on this issue. The in-situ high-

resolution STEM was indeed used under vacuum, but we have considered such different conditions and then carried out the sintering experiments in tube furnace under vacuum before in-situ STEM test. In vacuum, we observed similar inter-particle distance dependent sintering behavior in 5% H₂/Ar: the long inter-particle distance catalyst (10% Pt/ BP2000) exhibited a superior sintering-resistance, the short inter-particle distance catalyst (10% Pt/XC-72R) was prone to severe sintering by PMC path (Fig. R17), which was demonstrated by the formation the neck structure between two particles¹².

To exclude the possible “knock-on” damage at 200 kV, before individually tracking fixed area (Fig. 4a,b), we have also selected several other areas to compare the changes before and after heating (Supplementary Figs. 6 and 7), which were not disturbed by the “knock-on” damage from electron beams. Meanwhile, we supplemented the only time-resolved STEM experiments with only electron beam treatment for 30 min (Fig. R18 and Videos 1, 2) and then started heating at 900 °C. We also performed the only heating STEM experiments, where the electron beam was closed during heating and only open to observe changes after the end of heating (Fig. R19). Only the “knock-on” damage of electron beam cannot change the particle state, but the thermal energy can induce changes in particles. These results suggested that the long inter-particle distance sample was prone to Ostwald ripening via atom species only derived by the thermal energy force, rather than the “knock-on” damage from electron beam.

Figure R17. The sintering test of Pt/BP2000 and Pt/XC-72R in tube furnace under

vacuum condition, showing similar inter-particle distance behavior in 5% H₂/Ar.

Figure R18. Time-resolved STEM experiments with electron beam treatment for 30 min and then start heating at 900 °C.

Figure R19. In-situ heating STEM experiments, where the electron beam was closed during heating and only open to observe changes after the end of heating.

The above results and discussion have been added in the revised manuscript (Page 7 and 9) and Supplementary Information (Fig. S12 and Fig. S18-S19)

5. Again for Fig. 4d and 4e, the authors used the two-dimensional area to represent the three-dimensional particle size, which only works for isotropic particles and not the case here.

Response: We agree with the reviewer on this issue and have removed the figure and

relevant discussion from the revised manuscript.

6. For the dehydrogenation measurement, can the authors also provide the activity based on the active sites of Pt, like TOF, for both samples, instead of percentage conversion? That will present more useful information for comparison.

Response: Many thanks for the reviewer's comments on this issue. As suggested by the reviewer, we have provided the specific activity based on the active sites of Pt, including the initial and final TOF of C_3H_8 after long-term reaction (Fig. R20).

Figure R20. Normalized C_3H_8 conversion versus the time-on-stream for different inter-particle distance and the change of TOF of C_3H_8 after long time reaction.

The above results and discussion have been added in the revised manuscript (Page 11 and Fig. 6).

Reference

1. Hu SL, Li WX. Metal-support interaction controlled migration and coalescence of supported particles. *Sci. China Technol. Sc.* **62**, 762-772 (2019).
2. Campbell CT, Sellers JR. Anchored metal nanoparticles: Effects of support and size on their energy, sintering resistance and reactivity. *Faraday discuss.* **162**, 9-30 (2013).
3. Ouyang R, Liu JX, Li WX. Atomistic theory of Ostwald ripening and disintegration of supported metal particles under reaction conditions. *J. Am. Chem. Soc.* **135**, 1760-1771 (2013).
4. Dietze EM, Plessow PN, Studt F. Modeling the Size Dependency of the Stability of Metal Nanoparticles. *J. Phys. Chem. C* **123**, 25464-25469 (2019).
5. Ko M, Padgett E, Yarlagadda V, Kongkanand A, Muller DA. Revealing the Nanostructure of Mesoporous Fuel Cell Catalyst Supports for Durable, High-Power Performance. *J. Electrochem. Soc.* **168**, 024512 (2021).
6. Padgett E, *et al.* Connecting fuel cell catalyst nanostructure and accessibility using quantitative cryo-STEM tomography. *J. Electrochem. Soc.* **165**, F173 (2018).
7. Sattler JJ, Ruiz-Martinez J, Santillan-Jimenez E, Weckhuysen BM. Catalytic dehydrogenation of light alkanes on metals and metal oxides. *Chem. Rev.* **114**, 10613-10653 (2014).
8. Higgins D, *et al.* Development and simulation of sulfur-doped graphene supported platinum with exemplary stability and activity towards oxygen reduction. *Adv. Fun. Mater.* **24**, 4325-4336 (2014).
9. Pylypenko S, *et al.* Nitrogen: unraveling the secret to stable carbon-supported Pt-alloy electrocatalysts. *Energ. Environ. Sci.* **6**, 2957-2964 (2013).
10. Ciapina EG, Santos SF, Gonzalez ER. Electrochemical CO stripping on nanosized Pt surfaces in acid media: A review on the issue of peak multiplicity. *J. Electroanal. Chem.* **815**, 47-60 (2018).
11. Yan Q-Q, *et al.* Reversing the charge transfer between platinum and sulfur-doped carbon support for electrocatalytic hydrogen evolution. *Nat. Commun.* **10**, 1-9 (2019).
12. DeLaRiva AT, Hansen TW, Challa SR, Datye AK. In situ Transmission Electron Microscopy of catalyst sintering. *J. Catal.* **308**, 291-305 (2013).

REVIEWER COMMENTS

Reviewer #1 (Remarks to the Author):

Although my comments in the previous round are rather negative, the authors well compensate deficiencies of the original manuscript. With this efforts for revisions, the manuscript becomes scientifically well balanced. I change my mind to support publication of this nice work in Nat. Commun.

Reviewer #2 (Remarks to the Author):

The authors have taken care to address the concerns that have been expressed and thereby have greatly improved the manuscript.

My only concern that remains is the sintering at 500C during catalysis whereas sintering is very limited at 700C in H₂/Ar.

The authors mention different exposure times but these are in the same ballpark. There might be a role of carbon in facilitating sintering? I suggest to address some proposals for this surprising observation

Reviewer #3 (Remarks to the Author):

The authors have addressed the issues raised.

Reviewer #1 (Remarks to the Author):

Although my comments in the previous round are rather negative, the authors well compensate deficiencies of the original manuscript. With this efforts for revisions, the manuscript becomes scientifically well balanced. I change my mind to support publication of this nice work in Nat. Commun.

Response: We sincerely appreciate the reviewer for the positive comments.

Reviewer #2 (Remarks to the Author):

The authors have taken care to address the concerns that have been expressed and thereby have greatly improved the manuscript.

My only concern that remains is the sintering at 500C during catalysis whereas sintering is very limited at 700C in H₂/Ar.

The authors mention different exposure times but these are in the same ballpark. There might be a role of carbon in facilitating sintering? I suggest to address some proposals for this surprising observation.

Response: We sincerely appreciate the reviewer for the comments. As suggested by the reviewer, we have added some discussion for such observation in the revised manuscript (page 12):

“Note that the sintering of 8.0%Pt/KJ300J catalyst was very limited in H₂/Ar at 700 °C for a short duration of 120 min (Fig. 2), while the same catalyst suffered severe at 500 °C for a long duration of 500 min under the condition of realistic propane dehydrogenation reaction (Fig. 6d). Such different sintering phenomena between the sintering test and the realistic propane dehydrogenation reaction could be ascribed to varied sintering time and atmosphere. Particularly, the realistic propane dehydrogenation reaction that involves numerous adsorption and desorption steps towards various carbon-containing intermediates could has a profound influence on the sintering of supported Pt catalysts”

Reviewer #3 (Remarks to the Author):

The authors have addressed the issues raised.

Response: We sincerely appreciate the reviewer for the comments.